# In Vitro Susceptibility to Imipenem/Relebactam and Comparators in a Multicentre Collection of *Mycobacterium abscessus* Complex Isolates

**DOI:** 10.3390/antibiotics14070682

**Published:** 2025-07-05

**Authors:** Alejandro Seoane-Estévez, Pablo Aja-Macaya, Andrea Garcia-Pose, Paula López-Roa, Alba Ruedas-López, Verónica Gonzalez-Galán, Jaime Esteban, Jorge Arca-Suárez, Martín Pampín, Alejandro Beceiro, Marina Oviaño, Germán Bou

**Affiliations:** 1Servicio de Microbiología, Instituto de Investigación Biomédica de A Coruña (INIBIC), Complejo Hospitalario Universitario A Coruña, Sergas, Universidade da Coruña (UDC) As Xubias, 15006 A Coruña, Spain; alejandro.seoane.estevez@sergas.es (A.S.-E.); pablo.aja.macaya@sergas.es (P.A.-M.); andrea.garcia.pose@sergas.es (A.G.-P.); jorge.arca.suarez@sergas.es (J.A.-S.); martin.pampin.garcia@sergas.es (M.P.); alejandro.beceiro.casas@sergas.es (A.B.); marina.oviano.garcia@sergas.es (M.O.); 2CIBER de Enfermedades Infecciosas (CIBERINFEC, CB21/13/00055), ISCIII, 28029 Madrid, Spain; 3Instituto de Investigación Hospital 12 de Octubre (Imas12), 28041 Madrid, Spain; plroa@salud.madrid.org (P.L.-R.); alba.ruedas.lo@gmail.com (A.R.-L.); 4Servicio de Microbiología, Hospital Universitario 12 de Octubre, 28041 Madrid, Spain; 5Servicio de Microbiología, Hospital Universitario Virgen del Rocío, 41013 Sevilla, Spain; veronica.gonzalez.galan.sspa@juntadeandalucia.es; 6Departmento de Microbiología Clínica, Instituto de Investigación Sanitaria-Fundación Jiménez Díaz, 28040 Madrid, Spain; jesteban@fjd.es; 7Departamento de Fisioterapia, Medicina, y Ciencias Biomédicas, Universidad de A Coruña, 15006 A Coruña, Spain

**Keywords:** imipenem/relebactam, *Mycobacterium abscessus* complex, pulmonary infections, clinical microbiology

## Abstract

**Background and Objectives:** Infections caused by non-tuberculous mycobacteria (NTM), including *Mycobacterium abscessus* complex (MABc), are increasing globally and are notoriously difficult to treat due to the intrinsic resistance of these bacteria to many common antibiotics. The aims of this study were to demonstrate the in vitro activity of imipenem/relebactam against MABc clinical isolates and to determine any in vitro synergism between imipenem/relebactam and other antimicrobials. **Methods:** A nationwide collection of 175 MABc clinical respiratory isolates obtained from 24 hospitals in Spain (August 2022–April 2023) was studied. Fifteen different antimicrobial agents were comprised, including imipenem/relebactam. MICs were determined according to CLSI criteria, and the synergism studies were performed with the selected clinical isolates. **Results:** Of the 175 isolates obtained, 110 were identified as *M. abscessus* subsp. *abscessus* (62.9%), 51 as *M. abscessus* subsp. *massiliense* (29.1%), and 14 as *M. abscessus* subsp. *bolleti* (8%). The antibiotics yielding the highest susceptibility rates were tigecycline, eravacycline, and omadacycline (100%); followed by imipenem/relebactam and clofazimine (97.6%); and finally amikacin (94.6%). Only four isolates were resistant to imipenem/relebactam, three of which were further characterized by WGS, revealing MABc mutations in Bla_Mab_ as well as D,D- and L,D-transpeptidades and mspA porin, which may play an important role in reduced susceptibility to imipenem/relebactam, even though none were previously described or associated with resistance to β-lactams. **Conclusions:** Our data demonstrate that relebactam improved the anti-MABc activity of imipenem, representing a β-lactam for the treatment of MABc infections. Furthermore, imipenem/relebactam demonstrated in vitro synergism with other anti-MABc treatments, thus supporting its use as part of dual regimens.

## 1. Introduction

The *Mycobacterium abscessus* complex (MABc), which comprises three subspecies (*M. abscessus* subsp. *abscessus*, *M. abscessus* subsp. *Massiliense*, and *M. abscessus* subsp. *bolletii*) is a rapidly growing nontuberculous mycobacterium (NTM) involved in pulmonary infections, skin and soft tissue infections, bacteraemia, and other infections, affecting both inmunocompromised and inmunocompetent patients. MABc pulmonary disease is an emerging infection of particular concern in people with bronchiectasis and cystic fibrosis, and in some geographical areas, it is considered the second most frequent NTM [1]. The increasing prevalence of MABc pulmonary disease has recently drawn significant attention due to the paucity of effective treatment regimens, highlighting an urgent need for new therapeutic strategies. Multiple factors contribute to the clinical challenges posed by these infections: (1) frequent misidentification arising from genetic and phenotypic similarities among MABc subspecies and other non-tuberculous mycobacteria (NTM); (2) a marked global rise in the incidence and prevalence of MABc infections; (3) the absence of standardized, evidence-based treatment guidelines; and (4) extensive antimicrobial resistance resulting from both intrinsic and acquired mechanisms.

Notably, MABc isolates exhibit high levels of resistance to multiple antibiotics in both in vitro and in vivo settings, with poor concordance between in vitro susceptibility results and clinical outcomes [2]. This discordance further complicates the selection of effective treatment regimens. The most recent consensus guidelines from multiple professional societies recommend a two-phase therapeutic approach for MABc infections. The initial intensive phase involves the administration of intravenous agents such as amikacin, tigecycline, and imipenem. This is followed by a continuation phase incorporating inhaled amikacin and oral antibiotics, including macrolides, clofazimine, and linezolid [3]. This phased strategy aims to improve clinical outcomes while addressing the pathogen’s intrinsic resistance and the high likelihood of treatment failure with monotherapy.

Currently, there are no FDA-approved drugs specifically indicated for the treatment of MABc infections. Establishing the antimicrobial activity of candidate agents through standardized in vitro susceptibility testing is a critical step in generating reliable, clinically applicable data to guide therapy [4]. Treatment regimens are prolonged, often extending from 6 to 24 months, and overall therapeutic outcomes remain suboptimal—particularly in cases involving *M. abscessus* subsp. *abscessus*. Despite appropriate antibiotic therapy, sputum culture conversion rates are as low as 30%, and the cure rates for MABc pulmonary disease range from 30% to 50%, comparable to those reported for extensively drug-resistant tuberculosis, especially in infections caused by subsp. *abscessus* [5]. Moreover, the therapeutic window is frequently constrained by the development of severe drug-related toxicities, which further limit treatment tolerability and success.

Imipenem and cefoxitin are currently the only β-lactam antibiotics recommended in clinical guidelines for the treatment of MABc infections, despite evidence that other β-lactams also exhibit in vitro activity. This limited recommendation is largely attributable to the stability of imipenem and cefoxitin against the intrinsic class A broad-spectrum β-lactamase, Bla_Mab_, inactivates the majority of β-lactam agents. Nonetheless, imipenem alone often yields high minimum inhibitory concentrations (MICs), necessitating dose adjustments to achieve pharmacokinetic/pharmacodynamic (PK/PD) targets [6]. Several studies have indicated that Bla_Mab_ is not significantly inhibited by β-lactamase inhibitors, namely clavulanate, tazobactam, and sulbactam [7,8]. Relebactam, a non-β-lactam competitive inhibitor of Ambler class A and C β-lactamases, has recently been approved by the FDA and the EMA in combination with imipenem and cilastatin for the treatment of urinary tract and intra-abdominal infections (Recarbrio, MSD). Relebactam has previously shown in vitro activity that enhances the activity of imipenem against MABc isolates (one-two fold dilutions), although there is little data available to favor the selection of this drug for inclusion as part of currently recommended anti-MABc regimes [9]. It is important to highlight that, when combined with ertapenem, the addition of relebactam may decrease 32-times the carbapenem MIC [10].

The primary objective of this study was to evaluate the in vitro activity of the imipenem/relebactam combination against a large collection of clinical MABc isolates obtained from respiratory samples collected at tertiary care hospitals across Spain. This activity was compared to that of imipenem alone and to 13 other antibiotics with reported or potential efficacy against MABc. In light of the growing interest in β-lactam combination strategies to enhance antimycobacterial activities through synergistic mechanisms, we also assessed the in vitro effects of combining imipenem/relebactam with other antibiotics currently recommended for the treatment of MABc pulmonary infections, including clarithromycin, amikacin, and tigecycline. Additionally, we explored the potential synergistic activity of imipenem/relebactam in combination with meropenem and ceftaroline—two agents with limited but emerging data supporting their use against MABc—as possible components of novel therapeutic regimens.

## 2. Materials and Methods

### 2.1. Study Design and Bacterial Isolates

A prospective multicenter study was designed to collect MABc isolates from clinical respiratory specimens between August 2021 and April 2023. The study was supported by the Mycobacterial Infection Study Group (GEIM) of the Spanish Society of Infectious Diseases and Clinical Microbiology (SEIMC). Twenty-four hospitals in ten regions in Spain participated in the study by collecting nonduplicate MABc isolates during the study period.

The study included *Mycobacterium abscessus* complex (MABc) isolates obtained from the following representative respiratory specimens: (1) pulmonary samples such as bronchoalveolar lavage (BAL) fluid, sputum, lung biopsy tissue, and material from lung abscesses; and (2) samples from patients with confirmed pulmonary infection. Pulmonary infection was defined by the presence of clinical respiratory symptoms, radiographic findings such as nodular or cavitary opacities on chest radiograph, or high-resolution computed tomography (HRCT) evidence of bronchiectasis accompanied by multiple nodules, in the absence of alternative diagnoses. Isolates deemed to represent colonization rather than active infection were excluded from the study.

Bacterial identification at the subspecies level was performed in each participating center using reference genotypic methods, such as multilocus sequencing analysis (MLSA, considering *hsp65*, *rpoB*, and *secA* genes) and/or the GenoType NTM-DR assay (Bruker Daltonik, Bremen, Germany) [11]. Each center submitted their isolates to the coordinating center (Clinical Microbiology Department, Complejo Hospitalario Universitario La Coruña, Spain), where antibiotic susceptibility testing of all isolates was performed. The bacterial strains were stored at −70 °C and subcultured on a blood agar (Biomerieux, Marcy l’Etoile, France) at 30 °C before analysis.

### 2.2. Antimicrobial Susceptibility Testing

All isolates were tested against the following 15 antimicrobial agents: amoxicilin, cefuroxime, cefepime, ceftaroline, meropenem, imipenem, imipenem/relebactam, amikacin, moxifloxacin, clofazimine, linezolid, clarithromycin, tigecycline, omadacycline, and eravacycline. MICs were determined by a broth microdilution method in 96-well plates, according to the recommendations of the National Committee for Clinical Laboratory Standards (CLSI, M24, 2018) [12] (Appendix A). Clinical categorization was performed following CLSI guidelines.

The inocula were prepared from actively growing bacteria in 10 mL of a cation-adjusted Mueller Hinton broth (CAMHB), adjusted to a bacterial cell density of 10^6^ colony forming units (CFUs) per mL and diluted to a final inoculum of 5 × 10^4^ cfu/well. Standard reference powders of each antimicrobial were serially diluted 2-fold in 50 μL of CAMHB. The range of antibiotic concentrations was 0.12–128 mg/L for amoxicilin, cefuroxime, cefepime, ceftaroline, meropenem, imipenem, imipenem/relebactam, amikacin, clarithromycin, moxifloxacin, and linezolid. For clofazimine, tigecicline, omadacicline, and eravacicline, the antibiotic concentrations ranged from 0.03 to 32 mg/L. The antibiotics were added to the wells of the plates, and the prepared inocula were then added (the final volume in each well was 100 μL: 50 μL of bacterial suspension + 50 μL of antibiotic solution). The plates were incubated at 30 °C.

The plates were examined after incubation for 72 h. When no growth was observed in the control wells, the incubation was prolonged for 24–48 h. The plates were further incubated up to 14 days for clarithromycin-susceptible isolates to observe inducible clarithromycin resistance. The MICs were interpreted at 80% inhibition for linezolid and at 100% inhibition for all other antimicrobials. Quality control was performed using *M. abscessus* subs. *abscessus* (ATCC 19977) for imipenem, amikacin, claritromicin, moxifloxacin and linezolid as per expected MIC ranges recommended by CLSI (all were within the range). MICs for each antimicrobial agent were determined in duplicate. In cases where two different MIC values were obtained, the higher value was recorded.

For all antibiotics, both MIC_50_ and MIC_90_ values were determined. The MICs were interpreted according to CLSI criteria (see Appendix A). MICs of tigecycline and clofazimine were interpreted using the modified values by Petrini and Shen [13,14], respectively. Also, MIC_50_ and MIC_90_ values were determined for the most active antibiotics stratified by subspecies (Appendix A).

### 2.3. Dual-Drug (Based on Imipenem/Relebactam) Susceptibility Testing by the Checkerboard Method and FIC Determination

Once the susceptibility profile of all the MABc isolates was determined, a checkerboard microdilution assay was used to test the combinations of several antibiotics with imipenem/relebactam in vitro. Selection of the antibiotics to be tested in combination with imipenem/relebactam was based on the fact that all of them are intravenous drugs used in the intensive phase of treatment.

Checkerboard microdilution assays were performed in duplicate to evaluate the in vitro activity of imipenem/relebactam in combination with ceftaroline, meropenem, tigecycline, clarithromycin, and amikacin. Imipenem alone was included as a control. A total of 16 representative *M. abscessus* complex isolates were tested, including four isolates that demonstrated initial resistance to imipenem/relebactam. The isolates were selected on the basis of having a representative collection of the different geographical origins of the sample comprising a wide range of imipenem/relebactam MICs.

The concentration of imipenem/relebactam ranged from 0.12 to 128 mg/L, when combined with ceftaroline, meropenem, tigecycline, and amikacin, and from 1 to 64 mg/L when combined with clarithromycin. On the other hand, the range of antibiotic concentrations for meropenem and ceftaroline was 4–256 mg/L; 0.5–32 mg/L for amikacin; 0.06–4 mg/L for tigecycline; and 0.06–64 mg/L for clarithromycin. The concentrations of each antibiotic were prepared individually using CAMHB. The plates were then inoculated to yield a final volume of 100 μL per well (50 μL of each antibiotic). The bacterial inoculum was adjusted and distributed in all the wells to a final inoculum of 5 × 10^4^ cfu/well. In each plate, two wells were used as positive and negative controls and inoculated with only 100 μL of CAMHB (the positive control well was inoculated with the bacterial inoculum). The plates were incubated at 30 °C and examined after incubation for 2 h. The fractional inhibitory concentration index (FICI) was used to study the in vitro susceptibility relationship between two antibiotics. The FICI can be calculated as FIC_A_ + FIC_B_ = FICI, where FIC_A_ = MIC of drug A in combination/MIC of the drug A alone and FIC_B_ = MIC of the drug B in combination/MIC of the drug B alone. The FICI values were interpreted according to the CLSI Standards as follows: synergy if FICI ≤ 0.5, indifference if FICI is between 0.5 and 4, and antagonism if FICI > 4.

### 2.4. Whole Genome Sequencing

Three imipenem/relebactam resistant isolates (no imipenem MIC reduction in the presence of the inhibitor) (isolates number 16, 47, and 64) were studied by WGS in an attempt to detect novel resistance determining mutations, as the addition of relebactam did not improve the activity of imipenem. Additionally, two imipenem/relebactam susceptible isolates (MIC reduction in the presence of the inhibitor) (isolates number 17 and 46) were selected to undergo WGS for comparative purposes. These five isolates were sequenced in an Illumina MiSeq benchtop sequencer (Illumina, San Diego, CA, USA).

Briefly, total genomic DNA was obtained from MABc isolates grown from blood agar by using the QIAamp DNA Mini Kit (Qiagen, Hilden, Germany) according to the manufacturer’s instruction. Purified genomic DNA from all isolates was then sequenced. Quality control of paired-end Illumina reads was performed using fastp (v.0.32.2) [15] and then assembled with Unicycler (v.0.5.0) [16]. The integrity, contamination, and general quality of the generated assemblies were assessed using CheckM (v.1.1.3) [17] and identified with Kmerfinder (v.3.2) [18] and mlst (Center for Genomic Epidemiology, v2.0) [19]. Gene annotation was conducted using bakta (v.1.7.0) [20]; resistance genes were detected using RGI (v.5.2.0) and CARD (v.3.2.8) [21], and plasmids were detected using MOB-suite (v.3.1.0) [22]. Phylogenomic analyses were performed using Snippy (v.4.6.0) [23] and ggtree [24] with *M. abscessus* ATCC-19977 (NC_010397) as the reference strain.

Genes coding for MspA (MAB_4098), PpiA (MAB_2874), Bla_Mab_ (MAB_2875), PbpA (MAB_0035c), PonA2 (MAB_0408c), PonA1 (MAB_4901c), Ldt1 (MAB_3165), Ldt2 (MAB_1530), Ldt3 (MAB_4775), Ldt4 (MAB_4537c), and Ldt5 (MAB_4061) were analyzed in depth, comparing both imipenem/relebactam phenotypes. Additionally, other genes possibly involved in antibiotic resistance, peptidoglycan biosynthesis, and remodeling, as described by Rifat D. et al., were analyzed, and the data are shown in Appendix A [25].

## 3. Results

### 3.1. Distribution of the M. abscessus Complex in the Study

A total of 192 MABc isolates were initially collected, but 13 of these were excluded as they did not fulfill the bur criteria. The remaining 175 isolates included 39 BAL, 132 from sputum, 3 from lung biopsy, and 1 from lung abscess.

Of the 175 isolates, 110 were identified as *M. abscessus* subsp. *abscessus* (62.9%), 51 as *M. abscessus* subsp. *massiliense* (29.1%), and 14 as *M. abscessus* subsp. *bolleti* (8%).

### 3.2. Antimicrobial Susceptibility

Of the 175 MABc isolates tested, 8 were discarded, as no growth was observed on the plates on day five of incubation; thus, a total of 167 of MABc isolates were studied. All isolates (100%) were resistant to amoxicilin and cefuroxime (MICs > 128 mg/L). For the other antimicrobial agents, the distribution of MICs is outlined in Table 1 without differentiating the subspecies. We found no difference in MICs for *M. abscessus* subsp. *abscessus* and *M. abscessus* subsp. *massiliense* (except for the clarithromycin MIC). MIC_50_ and MIC_90_ remain almost invariable regardless of the subspecies for the most active antibiotics (Appendix A).

In general, tigecycline, clofazimine, and amikacin yielded the highest susceptibility rates of, respectively, 100%, 97.6%, and 94.6%. Linezolid exhibited a wide range of MICs (1–32 mg/L) with a global susceptibility rate of 58.1%. For clarithromycin, the resistance rate measured on day 3 was 3.6%, increasing to 51.5% when the incubation was prolonged up to day 14, highlighting inducible clarithromycin resistance. Almost all isolates were resistant to moxifloxacin. The MICs of eravacycline (≤1 mg/L) and omadacycline (≤2 mg/L) showed that 100% of the isolates were inhibited, confirming the potent in vitro activity of these agents. The good activity of omadacycline towards *M. abscessus* has been previously reported (Shoen C et al. 2019) [26].

Among β-lactams, the MICs of cefepime, ceftaroline, and meropenem ranged from 64 to >128, 16 to >128, and 16 to >128 mg/L, respectively. For imipenem, the MICs ranged from 2 to >32 mg/L with a resistance rate of 10.2%; combining imipenem with relebactam yielded good activity, reducing the resistance rate to 2.4% and indicating the good potential of this combination. For 120 isolates, the addition of relebactam yielded a 2-fold reduction in the imipenem MICs, and for 3 isolates, a 4-fold reduction was obtained. Interestingly, of the 17 isolates with an imipenem MIC of 32 mg/L (i.e., resistant to imipenem), only 4 isolates remained resistant (no reduction in the MIC despite the addition of relebactam; isolates number 8, 16, 47, and 64; Appendix A). The MIC_50_ and MIC_90_ values for imipenem/relebactam were 8 and 16 mg/L, respectively, while for imipenem alone, the MICs were 16 and 32 mg/L.

### 3.3. In Vitro Synergism of Combinations Based on Imipenem/Relebactam

Of the 16 MABc selected isolates, including the above-mentioned 4 imipenem/relebactam resistant isolates, the combinations exhibiting the strongest synergistic effects were imipenem/relebactam plus amikacin (87.5%, 14/16 isolates), imipenem/relebactam plus meropenem (75%, 12/16 isolates), and imipenem/relebactam plus ceftaroline (62.5%, 10/16 isolates). Imipenem/relebactam plus clarithromycin and imipenem/relebactam plus tigecycline showed a moderate synergistic effect (50%, 8/16 isolates for both combinations) (Table 2). The incubation of the plates containing imipenem/relebactam plus clarithromycin was not extended to day 14, as the stability of imipenem/relebactam can be affected by long incubation. Overall, we did not observe any antagonistic effects in the combinations of imipenem/relebactam with any of the other antimicrobial agents (Table 2). Of the 4 imipenem/relebactam-resistant isolates, synergy was only demonstrated for the combination of imipenem/relebactam plus amikacin in 2 isolates. For the other combinations, no synergy was observed in any of the 4 imipenem/relebactam-resistant isolates. In vitro synergism experiments were performed with imipenem alone as control.

### 3.4. β-Lactam Resistance Determinants

In order to clarify whether imipenem/relebactam resistance could be determined by mutations in Bla_Mab_ (MAB_2875), we performed a gene sequence comparison for all five isolates: #16, 47, and 64 (no MIC differences were observed between imipenem and imipenem/relebactam) and #17 and 46, in which a decrease in the MIC of imipenem/relebactam was observed compared to that of imipenem by WGS (Figure 1). The comparison of Bla_Mab_ between the former isolates analyzed revealed two different patterns of mutations. The first one was shared between 2 out of the 3 isolates (isolate 16 and 64), where the same mutations were found in 11 out of the 13 modifications analyzed in the resistant isolates. The other pattern is represented by isolate 47 and revealed only 1 mutation in the Bla_Mab_, which was not shared by any other isolate. Notably, 6 mutations in the Bla_Mab_ were also found in susceptible isolates, thus not being possible to associate them with this increase in imipenem/relebactam resistance. Therefore, only 7 mutations in Bla_Mab_ could possibly be related to this decrease in susceptibility to imipenem/relebactam: Ala6Thr, Gly26Asp, Ala36Thr, Leu86Gln, Arg201Gly, Thr268Ala, and Val283Ala. The magnitude of the MIC shift associated with these mutations is two dilutions above the MIC_50_ and one dilution above the MIC_90,_ suggesting a possible moderate effect in the Bla_Mab_ inhibition. This needs further site-directed mutagenesis for confirmation.

In addition to the β-lactamase activity, the binding affinity of D,D-transpeptidases (DDTs) and L,D-transpeptidases (LDTs) to β-lactams also limits the activity of β-lactams against MABc. Both LDTs and PBPs are critical to survival because they maintain the structure and rigidity of peptidoglycan. A recent study found the main targets of β-lactams in MABc: PonA1 (D,D-transpeptidase) (MAB_4901c), PonA2 (D,D-transpeptidase) (MAB_0408c), Poppa (D,D-transpeptidase) (MAB_0035c), and Ldt4 (L,D-transpeptidase) (MAB_4537c) [27]. Two different deletions were detected in Ldt4 (MAB_4537c, Figure 1 and Appendix A) between the susceptible isolates (17 and 46) and two of the resistant isolates (16 and 64). However, they resulted in exactly the same protein, causing a proline deletion in a homopolymeric proline region. Interestingly, the third isolate (number 47) had very few mutations in these genes (see Figure 1), despite being resistant to imipenem/relebactam. Additional proteins involved in β-lactam resistance are shown and included *ppiA* (gene preceding Bla_Mab_), Ldt 1 to 5, and *mspA* (porin). In all cases, a non-clear pattern of amino acid changes could be observed between both group of isolates. A full list of mutations in additional genes possibly related to the antibiotic resistance of the five isolates (#16, 47, 64, 17, and 46) is available in Appendix A.

## 4. Discussion

In this study, we assessed the activity of the β-lactamase inhibitor relebactam against *Mycobacterium abscessus* complex (MABc) isolates by evaluating its efficacy in combination with imipenem. Our findings demonstrate that the imipenem/relebactam combination exhibits enhanced in vitro activity compared to imipenem alone. These results are consistent with previously published data, which report 2- to 4-fold reductions in the minimum inhibitory concentration (MIC) of imipenem when combined with relebactam [10,28], supporting its potential role as an active intravenous agent during the intensive phase of MABc treatment. We have also demonstrated no difference regardless of the subspecies, thus being helpful even in settings with no reference microbiology laboratory. We also demonstrated the in vitro activity of omadacycline and eravacycline, two novel tetracycline derivatives, again confirming the findings of previous studies [29]. As in the case of imipenem/relebactam, these drugs may become part of anti-MABc regimes in the coming years.

Recent studies have suggested that combining β-lactams from different classes with other antibiotics may produce synergistic effects against MABc, likely due to the broader spectrum of bacterial targets engaged by such combinations [30]. In our study, we demonstrated that the combination of imipenem/relebactam with amikacin exhibits significant synergistic activity, including isolates initially resistant to imipenem alone.

When attempting to elucidate the mechanism of imipenem/relebactam resistance, we found 7 mutations in the Bla_Mab_ gene not previously associated with β-lactam resistance. Regarding other important genes (*ppiA*, *pbpA*, *pbpB*, *ponA1*, *ponA2*, *mspA*, and *ldt1* to *ldt5*), several mutations were observed, but their importance is unclear, and further research is needed. The ability of distinct MABc isolates to present different mutations on several PBPs may lead to present different susceptibility patterns. Based on WGS data obtained from these clinical isolates, we hypothesize that the production of Bla_Mab_ combined with mutations of several PBPs (*ponA1*, *ponA2*, *pbpA,* and *pbpB*) and L,D-transpeptidases could affect the imipenem/relebactam activity in MABc [31]. Further structural studies are required to elucidate the role of specific mutations, particularly to clarify the impact of penicillin-binding proteins (PBPs) on reduced susceptibility to the imipenem/relebactam combination (IMI/REL). Functional validation approaches, such as site-directed mutagenesis or transcriptomic profiling, are necessary to confirm these findings. Additionally, phenotypic associations of mutations are limited by the considerable phylogenetic diversity among isolates, which complicates the identification of resistance-associated variants, as many mutations may be unrelated due to their presence in different sequence types (STs).

Evolution towards high levels of resistance to imipenem in *M. abscessus* has been described [32]. Resistance to imipenem was associated with increased β-lactamase activity and increased levels of Bla_Mab_ mRNA. Deletion of the *mspA* porin coincided with the first increase in MIC (from 8 to 32 mg/L). Subsequent mutations in *msp2* and *hrpA* coincided with a second increase in MIC (from 32 to 256 mg/L). None of these changes were observed in strains with reduced sensitivity to imipenem/relebactam, although an increase in the blaMAB expression levels cannot be ruled out. It is necessary to highlight the work of Dousa Km et al. 2020, who conducted an extensive analysis of the potential role of relebactam in inhibiting BlaMab [33]. Kinetic data and docking studies in this paper demonstrated that the larger aromatic piperidine group of relebactam interacts with the phenylalanine residue F237 of BlaMab. The study also showed that relebactam is a less potent inhibitor of blaMab, with a relatively poor affinity compared to avibactam.

Several studies have found no additional benefit in adding a β-lactamase inhibitor (BLI) to carbapenems, a phenomenon that is not fully understood. Even with a more potent inhibitor like durlobactam, the addition of durlobactam to imipenem failed to demonstrate a significant improvement in MICs, as stated previously [33].

The rationale for adding relebactam to imipenem to enhance susceptibility remains an open question. Interestingly, relebactam in combination with imipenem has been shown to protect amoxicillin from hydrolysis by BlaMab [28], suggesting that significant β-lactamase inhibition may occur when an appropriate substrate is present. Our study demonstrates that certain β-lactams, notably imipenem, exhibit improved activity against MABc isolates in the presence of relebactam, as evidenced by a one-dilution reduction in the MIC_50_, MIC_90_, and MIC range limits of imipenem by one dilution. Given that imipenem MICs against clinical MABc isolates commonly hover near the recommended susceptibility breakpoints, the imipenem/relebactam combination is expected to enhance pharmacokinetic/pharmacodynamic (PK/PD) target attainment, thereby increasing the efficacy relative to imipenem monotherapy. These findings are fully consistent with those reported by Kaushik et al. [10], who observed reductions of one- to three two-fold dilutions in MIC_50_ and MIC_90_ values for imipenem and meropenem, respectively, when combined with relebactam. Notably, a greater reduction in MIC values was observed with meropenem in the presence of the β-lactamase inhibitor compared to imipenem, suggesting the potential utility of regimens combining imipenem/relebactam with meropenem for treating MABc infections. Supporting this hypothesis, a synergy between meropenem and imipenem/relebactam was observed in 12 of the 16 MABc clinical isolates tested (Table 2). A similar study published by Burke A et al. also concluded that the MIC of imipenem/relebactam was one-fold dilution less than that of imipenem alone, with the same values for imipenem and imipenem/relebactam MIC_50_ and MIC_90_, reinforcing the value of our results [34]. In this case, no synergy studies were performed.

Regarding the treatment regimen for MABc infections and potential synergistic effects, the study by Story-Roller et al. provides important insights by testing paired antibiotic combinations for in vitro synergy against *M. abscessus* [35]. Their screening of 206 antibiotic pairs identified 24 combinations exhibiting synergy, including dual β-lactams, β-lactam plus β-lactamase inhibitor, and β-lactam plus rifamycin pairs. Consistent with these findings, our results demonstrate that imipenem/relebactam, combined with meropenem, ceftaroline, amikacin, clarithromycin, or tigecycline, exhibited a synergistic activity against a significant proportion of the 16 clinical MABc isolates tested. These data highlight the potential utility of such combinations in the management of chronic and recalcitrant pulmonary MABc infections. Furthermore, a recent study by Bitar et al. aligns with our observations, reporting that the amoxicillin/imipenem/relebactam combination not only demonstrated in vitro synergy but was also effective in vivo against MABc infections, underscoring the potential clinical benefit of these combination therapies [36].

## 5. Conclusions

The introduction of new drugs and combinations of drugs for the treatment of MABc pulmonary infections should be accompanied by extensive studies of both their in vitro activity and of their correlation to in vivo outcomes, based on well-controlled clinical trials for MABc infection. The main limitation of this study is precisely the fact that we did not study the correlation between the in vitro and in vivo results. Future investigations assessing the in vivo activity of imipenem/relebactam alone and in combination regimens, including dual-β-lactam therapies, are warranted to optimize dosing strategies and therapeutic protocols.

As far as we are aware, this is the largest study investigating the activity of imipenem/relebactam against clinical MABc isolates. The data presented herein demonstrate that relebactam improves the anti-MABc activity of imipenem, representing a β-lactam for the treatment of MABc infections, particularly when combined with amikacyn. Hence, we propose the use of imipenem/relebactam, alone or in combination, as a suitable treatment in MABc infections.

## Figures and Tables

**Figure 1 antibiotics-14-00682-f001:**
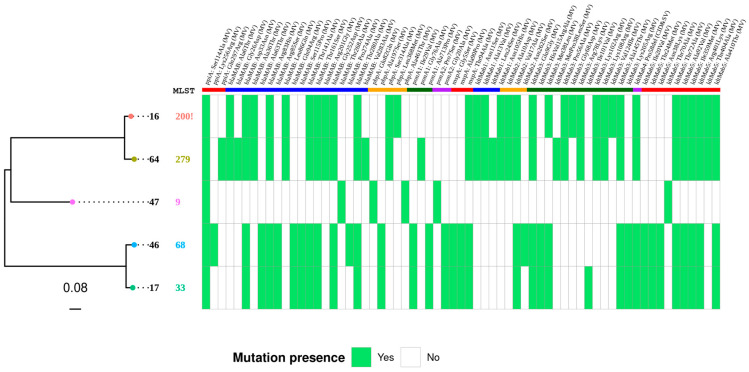
Core genome phylogenetic tree of the MABc-sequenced strains. Isolates 46 and 17 were susceptible, and 16, 64, and 47 were resistant to imipenem/relebactam. The figure represents, from left to right, the strain ID, the MLST, and the variant calling of genes *ppiA*, *bla_MAB_*, *pbpA*, *ponA1*, *ponA2*, *mspA*, *ldt1*, *ldt2*, *ldt3*, *ldt4,* and *ldt5*. An exclamation mark next to the ST indicates the closest described ST. Mutation significance is indicated through MV (missense variant), CID (conservative in-frame deletion), SV (synonymous-variant), and DID (disruptive in-frame deletion).

**Table 1 antibiotics-14-00682-t001:** Minimum inhibitory concentration (MIC) distribution, MIC_50_, and MIC_90_ among the 175 MABc isolates tested in the study.

MIC Distribution for MABc Isolates (n = 175)															MIC_50_	MIC_90_	%R
	≤0.03	0.06	0.12	0.25	0.5	1	2	4	8	16	32	64	128	>128			
Cefepime												3	18	146	>128	>128	^c^
Ceftaroline										1	2	14	33	117	>128	>128	^c^
Meropenem										5	44	77	28	13	64	128	^c^
Imipenem							1	19	58	72	17				16	32	10.2
Imipenem/relebactam						1	10	48	79	25	4				8	16	2.4
Linezolid						14	27	18	38	63	7				8	16	4.2
Moxifloxacine						3	10	51	62	41					8	16	92.2
Clarithromycin 72 h ^a^			73	35	28	18	5	2		2	4				0.25	1	3.6
Clarithromycin 14d ^b^			25	24	16	12	4				72	14			32	32	51.5
Omadacycline			3	14	49	66	35								1	2	^c^
Eravacycline		22	42	49	37	17									0.25	1	^c^
Amikacine					1	4	16	42	67	28	7	2			8	16	1.2
Tigecyclina	1	2	21	24	35	51	31	2							0.5	2	0
Clofazimine		3	24	63	45	28	3	1							0.25	1	0.6

^a^ MICs were measured at 72 h; ^b^ MICs were measured at 14 days; ^c^ Clinical categorization is not possible as there are no available breakpoints.

**Table 2 antibiotics-14-00682-t002:** In vitro synergism of combinations based on imipenem/relebactam plus meropenem, ceftaroline, amikacin, clarithromycin, and tigecycline against a series of MABc isolates.

	IMI/REL—Meropenem					IMI/REL—Ceftaroline					IMI/REL—Amikacin					IMI/REL—Clarithromycin					IMI/REL—Tigecycline				
Isolate ID	A	a	B	b	FICI	A	a	B	b	FICI	A	a	B	b	FICI	A	a	B	b	FICI	A	a	B	b	FICI
8	32	16	128	32	0.75	32	16	128	64	1	32	16	8	2	0.75	32	16	0.3	0.1	1	32	8	0.1	0.1	0.75
16	32	16	128	64	1	32	16	128	64	1	32	16	8	4	1	32	16	32	16	1	32	16	0.3	0.1	0.8
17	8	2	64	8	0.38	8	2	64	8	0.38	8	2	16	4	0.5	8	4	32	4	0.6	8	4	0.5	0.3	1
29	16	4	32	8	0.5	16	4	128	16	0.38	16	4	16	2	0.38	16	4	64	16	0.5	16	4	0.1	0	0.37
47	32	16	128	64	1	32	16	128	16	0.63	32	8	16	4	0.5	32	16	32	8	0.8	32	16	1	0.5	1
56	2	0.5	32	2	0.31	2	0.5	128	16	0.38	2	0.5	4	0.5	0.38	2	0.5	32	8	0.5	2	0.5	2	0.5	0.5
62	8	2	64	16	0.5	8	2	64	16	0.5	8	1	64	4	0.19	8	2	32	8	0.5	8	2	2	0.5	0.5
64	32	16	128	32	0.75	32	16	128	32	0.75	32	8	4	0.5	0.38	32	16	32	16	1	32	8	1	0.5	0.75
75	16	4	64	8	0.38	16	4	64	16	0.5	16	4	32	1	0.28	16	4	64	16	0.5	16	4	2	0.5	0.5
90	8	2	128	32	0.5	8	2	64	8	0.38	8	2	8	1	0.38	8	2	64	4	0.3	8	2	2	0.5	0.5
110	8	2	64	8	0.38	8	2	64	8	0.38	8	2	4	1	0.5	8	4	2	0.5	0.8	8	4	2	1	1
121	4	1	64	8	0.38	4	1	32	8	0.5	4	1	8	1	0.38	4	1	32	4	0.4	4	1	1	0.1	0.37
137	16	4	128	32	0.5	16	8	64	16	0.75	16	4	2	0.5	0.5	16	8	32	16	1	16	8	1	0.5	1
145	4	1	32	8	0.5	4	1	128	32	0.5	4	1	32	4	0.38	4	1	32	8	0.5	4	2	2	1	1
158	8	1	128	16	0.25	8	4	64	16	0.75	8	1	8	1	0.25	8	2	2	0.25	0.4	8	2	0.5	0.1	0.37
162	16	4	64	8	0.38	16	4	64	4	0.31	16	4	4	0.5	0.38	16	8	1	0.5	1	16	4	1	0.1	0.31
	**IMI—Meropenem**					**IMI—Ceftaroline**					**IMI—Amikacin**					**IMI—Clarithromycin**					**IMI—Tigecycline**				
**Isolate ID**	**A**	**a**	**B**	**b**	**FICI**	**A**	**a**	**B**	**b**	**FICI**	**A**	**a**	**B**	**b**	**FICI**	**A**	**a**	**B**	**b**	**FICI**	**A**	**a**	**B**	**b**	**FICI**
8	32	16	128	16	0.63	32	16	128	64	1	32	16	8	2	0.8	32	8	0.3	0.1	0.8	32	16	0.1	0	0.75
16	32	16	128	16	0.63	32	16	128	64	1	32	32	8	4	1.5	32	4	32	32	1.13	32	16	0.3	0.1	0.74
17	16	8	64	16	0.8	16	16	64	2	1.03	16	8	16	4	0.8	16	8	32	8	0.8	16	8	0.5	0.1	0.74
29	16	16	32	4	1.13	16	16	128	4	1.03	16	4	16	8	0.8	16	4	64	32	0.8	16	16	0.1	0	1.13
47	32	16	128	64	1	32	16	128	64	1	32	32	16	2	1.13	32	8	32	16	0.8	32	32	1	0.3	1.25
56	8	4	32	16	1	8	2	128	64	0.8	8	8	4	0.3	1.06	8	1	32	32	1.13	8	8	2	0.3	1.13
62	8	4	64	32	1	8	8	64	2	1.03	8	8	64	4	1.06	8	1	32	32	1.13	8	8	2	0.1	1.06
64	32	16	128	16	0.63	32	32	128	4	1.03	32	32	4	0.5	1.13	32	16	32	16	1	32	32	1	0.1	1.06
75	32	16	64	8	0.63	32	32	64	2	1.03	32	16	32	4	0.63	32	16	64	32	1	32	16	2	0.5	0.75
90	16	8	128	32	0.8	16	16	64	2	1.03	16	16	8	0.5	1.06	16	8	64	8	0.63	16	16	2	0.3	1.13
110	16	8	64	16	0.8	16	16	64	2	1.03	16	16	4	0.5	1.13	16	2	2	2	1.13	16	16	2	0.3	1.13
121	8	2	64	32	0.8	8	8	32	1	1.03	8	8	8	1	1.13	8	4	32	4	0.63	8	8	1	0.1	1.12
137	32	32	128	8	1.06	32	32	64	2	1.03	32	32	2	0.3	1.13	32	4	32	32	1.13	32	32	1	0.1	1.12
145	8	4	32	8	0.8	8	8	128	4	1.03	8	8	32	4	1.13	8	2	32	16	0.8	8	8	2	0.3	1.13
158	16	8	128	32	0.8	16	16	64	2	1.03	16	16	8	0.5	1.06	16	8	2	0.5	0.8	16	16	0.5	0	1.06
162	16	8	64	32	1	16	16	64	2	1.03	16	16	4	0.1	1.03	16	2	1	1	1.13	16	8	1	0.5	1

The following are represented for each isolate: A, MIC of IMI/REL (above) or imipenem (IMI) tested alone (below); a, MIC of IMI/REL or IMI tested in combination; B, MIC of the other antibiotic tested alone; b, MIC of the other antibiotic tested in combination with IMI/REL (above) or IMI (below); FICI, fractional inhibitory concentration index. Tables above: combinations with IMI/REL. Tables below: combinations with IMI alone as a control.

## Data Availability

The WGS data of the strains analyzed have been deposited in the GenBank database PRJNA1177655.

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
