# Peer review of "In Vitro Susceptibility to Imipenem/Relebactam and Comparators in a Multicentre Collection of Mycobacterium abscessus Complex Isolates"

_antibiotics, 2025, doi:10.3390/antibiotics14070682_

Round 1

Reviewer 1 Report

Comments and Suggestions for Authors

The focus of the current study is to analyze the “In-vitro susceptibility to imipenem/relebactam and comparators in a multicentre collection of Mycobacterium abscessus complex isolates”. The overall work done appears to be fine and correlates with the objective of the study. However, there are  comments to address as follows:

1. The correlation analysis of the observations of in-vitro results with in-vivo results is not performed.

2. The MICs are not performed in triplicates.

3. The inclusion criteria of the study should be provided in methodology.

4. Some recent similar work, like Burke et al, 2023 “In-vitro susceptibility testing of imipenem-relebactam and tedizolid against 102 Mycobacterium abscessus isolates” has been cited and the author should discuss the observations and differences in comparison this current study.

5. The English quality can be improved.

6. There are typographical errors. Italics are missing wherever required like invitro etc. The difference in points and comas are not clear in many places. Similar errors are visible in data provided.

Author Response

The focus of the current study is to analyze the “In-vitro susceptibility to imipenem/relebactam and comparators in a multicentre collection of Mycobacterium abscessus complex isolates”. The overall work done appears to be fine and correlates with the objective of the study. However, there are comments to address as follows:

  1. The correlation analysis of the observations of in-vitroresults with in-vivo results is not performed.

We really thank the reviewer for the time and comments on the paper. As noticed by the reviewer this correlation has not been explored in this paper. However, we have already acknowledged this limitation in the original draft (conclusions section). We think the in-vivo experiments are necessary, but they really deserve another complete study and it is far beyond the objectives of the present one.

  1. The MICs are not performed in triplicates.

We again have to acknowledge this limitation, however, as detailed in lines 148-152, MICs were performed in duplicates with a suitable quality control (M. abscessus subs. Abscessus, ATCC 19977) and following CLSI guidelines. Initially MICs were performed in triplicates, but as results were replicable, we decided to finally perform duplicates in order to ease the procedure.

  1. The inclusion criteria of the study should be provided in methodology.

The criteria is already detailed in 2.1 Study design and bacterial isolates (lines 123-131).

  1. Some recent similar work, like Burke et al, 2023 “In-vitrosusceptibility testing of imipenem-relebactam and tedizolid against 102 Mycobacterium abscessus isolates” has been cited and the author should discuss the observations and differences in comparison this current study.

This paper was not cited in the original draft, however as recommended by the reviewer we decided to include and discuss this work in lines 384-388, cite 36. We really think this work is line with our results reinforcing the value of the prescription of imipenem/relebactam for MABc infections

  1. The English quality can be improved.

We have carefully revised the manuscript so the language is improved. I hope it meets the expectations of the reviewer and the journal.

  1. There are typographical errors. Italics are missing wherever required like invitro etc. The difference in points and comas are not clear in many places. Similar errors are visible in data provided.

We have carefully revised the manuscript so these errors are missing from the current version.

Reviewer 2 Report

Comments and Suggestions for Authors
  1. Adequate sample size and wide sources: This study included 175 clinical respiratory isolates from 24 hospitals in Spain, with strong representativeness and multi-center credibility.
  2. Method specification, technical standards.
  3. This study tested a comprehensive antimicrobial spectrum and compared a total of 15 antimicrobial drugs.
  4. This study is the first to systematically analyze the synergistic effect of imipenem/relebactam in M. abscessus complex and explore the preliminary resistance mechanism, which is novel.

Disadvantages: 1. The study screened 175 clinical isolates for MIC testing, but only 16 strains were selected for drug synergy analysis, lacking statistical basis or explanation of screening criteria. Lack of representative analysis.

  1. Only 5 strains were selected for whole genome sequencing (WGS), of which half were resistant and half were sensitive (3 vs 2). The sample size was too small to draw a clear conclusion about the resistance mechanism. At the same time, no functional validation experiments were performed.
  2. Table 2. Are there other forms of presentation to facilitate readers' clear understanding?
  3. The reference to the anti-MABc study of relebactam in the introduction is relatively old (part of which is a previous in vitro study). It is recommended to supplement the systematic review literature on imipenem β-lactamase inhibition strategy in the past 2-3 years to enhance the argument support.

Author Response

Reviewer 2:

  1. Adequate sample size and wide sources: This study included 175 clinical respiratory isolates from 24 hospitals in Spain, with strong representativeness and multi-center credibility.
  2. Method specification, technical standards.
  3. This study tested a comprehensive antimicrobial spectrum and compared a total of 15 antimicrobial drugs.
  4. This study is the first to systematically analyze the synergistic effect of imipenem/relebactam in M. abscessus complex and explore the preliminary resistance mechanism, which is novel.

 We really would like to thank the reviewer for the time in reviewing the manuscript and for the nice appreciations performed. We have tried to improve the manuscript, following your recommendations. I hope it meets your expectations.

Disadvantages:

  1. The study screened 175 clinical isolates for MIC testing, but only 16 strains were selected for drug synergy analysis, lacking statistical basis or explanation of screening criteria. Lack of representative analysis.

We agree with the reviewer in the idea of having a lower sample size in the synergy testing compared with the individual testing of the isolates. However, due to the complexity of these studies, a representative sample collection was chosen. For that, we included the isolates with an impenem/relebactam MIC> 16 and then a representative collection of the different greographical origins of the sample comprising a wide range of imipenem/relebactam MICs. We have clarified it in the text (lines 183-185).

  1. Only 5 strains were selected for whole genome sequencing (WGS), of which half were resistant and half were sensitive (3 vs 2). The sample size was too small to draw a clear conclusion about the resistance mechanism. At the same time, no functional validation experiments were performed.

We have tried to make clear in the discussion section that a more robust analysis is needed to have solid conclusions and we are only hypothesizing based on our limited results (lines 368-375). We also discussed our results versus other similar works in the rest of the discussion section so that our assumptions are better delineated.

  1. Table 2. Are there other forms of presentation to facilitate readers' clear understanding?

We have tried to compress the tables and present them in a different format, so it is easier to compare results between the combinations of imipenem/relebactam and imipenem with a second antibiotic. I hope the reviewer find it easier as well.

  1. The reference to the anti-MABc study of relebactam in the introduction is relatively old (part of which is a previous in vitro study). It is recommended to supplement the systematic review literature on imipenem β-lactamase inhibition strategy in the past 2-3 years to enhance the argument support.

We have included a reference in the discussion section (36), also recommended by reviewer 1, in order to complete the discussion of the activity of imipenem/relebactam. Also, we have included a very recent reference (38) to support our conclusions and linked it in vivo experiments. I hope it improves the message in line of the reviewer comment (lines 409-413).

Reviewer 3 Report

Comments and Suggestions for Authors

This manuscript presents a comprehensive multicenter evaluation of the in vitro activity of imipenem/relebactam against a large panel of Mycobacterium abscessus complex (MABc) isolates collected from 24 hospitals in Spain. The study is well-conducted, methodologically sound, and timely given the growing clinical concern surrounding MABc infections and their limited treatment options. The inclusion of whole genome sequencing (WGS) and synergism assays adds valuable mechanistic and therapeutic insight.
Major Strengths
1) Large Isolate Collection: The use of 175 clinical isolates across different MABc subspecies strengthens the generalizability of the findings.
2) Well-Designed Susceptibility Testing: MIC testing adheres to CLSI guidelines, and the checkerboard method is appropriate for assessing synergy.
3) Integration of Genomic Analysis: WGS of both susceptible and resistant isolates is a significant strength that deepens the findings and supports exploratory associations with resistance mechanisms.
4) Novel Insights on Synergy: The demonstration of synergy between imipenem/relebactam and key agents such as amikacin and ceftaroline is clinically meaningful and may inform future combination therapies.

Recommendations

1.
The manuscript repeatedly references the potential clinical utility of imipenem/relebactam, but there is no in vivo validation or correlation with clinical outcomes. The authors should more clearly acknowledge this limitation and propose a pathway for validating these findings in animal models or clinical cohorts.

  1. The WGS findings suggest associations between certain mutations and resistance, but the causality remains speculative. Authors should temper their conclusions or clearly state that these mutations are hypothetically associated and require confirmation through functional validation (e.g., site-directed mutagenesis or transcriptomic profiling).
  2. The MIC data is not stratified by subspecies (abscessus, massiliense, bolletii) in most parts of the analysis. Consider including supplementary tables or a short section summarizing differences in susceptibility patterns across subspecies, as this has therapeutic implications.

    Overall, this is an important and well-conceived study that meaningfully contributes to the understanding of β-lactam/β-lactamase inhibitor combinations in MABc. While largely suitable for publication, minor revisions and clarifications, particularly in the genomic interpretation and data presentation, would improve the manuscript's impact and clarity.

Author Response

Reviewer 3:

This manuscript presents a comprehensive multicenter evaluation of the in vitro activity of imipenem/relebactam against a large panel of Mycobacterium abscessus complex (MABc) isolates collected from 24 hospitals in Spain. The study is well-conducted, methodologically sound, and timely given the growing clinical concern surrounding MABc infections and their limited treatment options. The inclusion of whole genome sequencing (WGS) and synergism assays adds valuable mechanistic and therapeutic insight.
Major Strengths
1) Large Isolate Collection: The use of 175 clinical isolates across different MABc subspecies strengthens the generalizability of the findings.
2) Well-Designed Susceptibility Testing: MIC testing adheres to CLSI guidelines, and the checkerboard method is appropriate for assessing synergy.
3) Integration of Genomic Analysis: WGS of both susceptible and resistant isolates is a significant strength that deepens the findings and supports exploratory associations with resistance mechanisms.
4) Novel Insights on Synergy: The demonstration of synergy between imipenem/relebactam and key agents such as amikacin and ceftaroline is clinically meaningful and may inform future combination therapies.

We really would like to thank the reviewer for the time in reviewing the manuscript and for the nice appreciations performed. We have tried to improve the manuscript, following your recommendations. I hope it meets your expectations.

Recommendations

1. The manuscript repeatedly references the potential clinical utility of imipenem/relebactam, but there is no in vivo validation or correlation with clinical outcomes. The authors should more clearly acknowledge this limitation and propose a pathway for validating these findings in animal models or clinical cohorts.

We have tried to make clear that in vivo experiments are needed to draw extensive conclusions. For that, in the new version of the manuscript, a conclusions section is included to address this limitation.

  1. The WGS findings suggest associations between certain mutations and resistance, but the causality remains speculative. Authors should temper their conclusions or clearly state that these mutations are hypothetically associated and require confirmation through functional validation (e.g., site-directed mutagenesis or transcriptomic profiling).

We agree with the reviewer. We have tried to clarify that we are only hypothesizing based on our limited results However, we have tried to emphasize this limitation in this new version of the manuscript following the reviewer recommendations (lines 368-375).

  1. The MIC data is not stratified by subspecies (abscessus, massiliense, bolletii) in most parts of the analysis. Consider including supplementary tables or a short section summarizing differences in susceptibility patterns across subspecies, as this has therapeutic implications.

As recommended by the reviewer we have stratified MIC results by subspecies. Although we have not seen important differences, we recognize the value of the analysis. Thus, we have included a new table S3 and a comment in the material and methods (lines 170-171), results section (lines 242-244) and discussion section (349-350).

Overall, this is an important and well-conceived study that meaningfully contributes to the understanding of β-lactam/β-lactamase inhibitor combinations in MABc. While largely suitable for publication, minor revisions and clarifications, particularly in the genomic interpretation and data presentation, would improve the manuscript's impact and clarity.

We have tried to improve the manuscript following your indications. I hope all issues have been clarified and the final draft meets your expectations.